# The Relationship between Gut Microbiota and Respiratory Tract Infections in Childhood: A Narrative Review

**DOI:** 10.3390/nu14142992

**Published:** 2022-07-21

**Authors:** Daniele Zama, Camilla Totaro, Lorenzo Biscardi, Alessandro Rocca, Silvia Turroni, Patrizia Brigidi, Marcello Lanari

**Affiliations:** 1Paediatric Emergency Unit, IRCCS Ospedale Maggiore Policlinico Sant’Orsola, Department of Medicine and Surgery, University of Bologna, 40138 Bologna, Italy; daniele.zama2@unibo.it (D.Z.); alessandro.rocca4@unibo.it (A.R.); marcello.lanari@unibo.it (M.L.); 2Specialty School of Pediatrics, Alma Mater Studiorum, University of Bologna, 40126 Bologna, Italy; camilla.totaro@studio.unibo.it; 3Unit of Microbiome Science and Biotechnology, Department of Pharmacy and Biotechnology, University of Bologna, 40126 Bologna, Italy; silvia.turroni@unibo.it (S.T.); patrizia.brigidi@unibo.it (P.B.)

**Keywords:** gut microbiota, gut–lung axis, respiratory tract infections, RTIs, childhood, children

## Abstract

Respiratory tract infections (RTIs) are common in childhood and represent one of the main causes of hospitalization in this population. In recent years, many studies have described the association between gut microbiota (GM) composition and RTIs in animal models. In particular, the “inter-talk” between GM and the immune system has recently been unveiled. However, the role of GM in human, and especially infantile, RTIs has not yet been fully established. In this narrative review we provide an up-to-date overview of the physiological pathways that explain how the GM shapes the immune system, potentially influencing the response to common childhood respiratory viral infections and compare studies analysing the relationship between GM composition and RTIs in children. Most studies provide evidence of GM dysbiosis, but it is not yet possible to identify a distinct bacterial signature associated with RTI predisposition. A better understanding of GM involvement in RTIs could lead to innovative integrated GM-based strategies for the prevention and treatment of RTIs in the paediatric population.

## 1. Introduction

RTIs are a major cause of morbidity and mortality worldwide. These infections can range from mild upper respiratory tract infections to severe and life-threatening pneumonia. Every year, more than 2 million children younger than 5 years die from pneumonia, accounting for nearly 20% of all deaths within this age group [1].

RTIs are divided into either viral or bacterial origin. Viral aetiologies have been documented in up to 80% of lower respiratory infections in children younger than 2 years [2]. Among viral pathogens, respiratory syncytial virus (RSV) is the most detected in children under 2 years of age [3]. RSV bronchiolitis is the leading cause of hospitalization in infants during the first year of life and one of the main causes of visits in paediatric emergencies [4].

Recurrent RTIs (RRTIs), defined as eight or more documented RTIs per year up to 3 years of age and six or more episodes in children older than 3 years, are also a very common clinical condition in pediatric age with an important social and economic impact [5,6,7,8]. In Italy, it is estimated that approximately 25% of children under 1 year of age and 6% of children during the first 6 years of life experience RRTIs, making them one of the main reasons for patient referral during the first years of life [6,9,10].

In recent years, a plethora of studies have described the relationship between GM composition and function and the development of various human diseases (Human Microbiome Project, https://hmpdacc.org/, accessed on 15 May 2022), focusing especially on the role of GM in regulating the immune system. Although the underlying mechanisms are not yet fully understood, there is mounting evidence that GM can modulate the immune function in distant mucosal sites such as the respiratory system, and therefore play a role in the development of RTIs.

This link has recently been defined as the gut–lung axis [11]. Data from animal models also suggest that GM dysbiosis may influence the severity of RTIs [12,13,14,15]. On the other hand, viral infections can alter the GM composition, as described in the murine model for influenza and RSV infection [16]. The relationship between GM and RTI therefore appears to be two-way, and it is not entirely clear whether dysbiosis is the cause or the consequence. In addition, a specific association between GM dysbiosis and RTIs in humans has not yet been clearly identified.

Specific differences between the immune response in adults and children within the context of both viral and bacterial infections exist, but not all pathways are fully elucidated [17,18,19]. During the global pandemic of SARS-CoV2, an important discrepancy appeared in the course of the disease among children and adults. This gap might be explained by various mechanisms, that will reassume some of the characteristics of viral immune responses in children: a stronger innate immune response of childhood [17], an immunesenescence in old people that leads to a suppression of adaptative immunity [17], and different activations of the immune response; a prominent Th1 was reported in SARS-CoV2-infected adults compared to children. [17]

Differences are observed also in bacterial infections. For example, as concerns H. pylori infection, the gastric concentrations of Th1 cytokines were lower in infected children than in adults and this occurrence coincides with a higher protection against the disease [18]. Whereas for the *Streptococcus pyogenes* infection, lower levels of IgG3 and IFN-γ were found in children compared with adults, reflecting a minor capability of children to respond to this infection or a history of numerous encounters with the bacterium and a subsequent better response in adulthood [19]. Considering the impact of RTIs in the pediatric population, the aim of this narrative review is to summarize and discuss the current knowledge about the association between GM and RTIs in children and the related clinical data.

## 2. Preclinical Data on the Gut–Lung Axis in RTIs

The concept that GM is vital not only for the gastrointestinal tract, but also for the overall health of the human organism, has been well studied over the past decade and is widely accepted today [11,20,21,22,23,24]. On the other hand, the presence of microbes in body niches previously thought to be sterile, such as the lungs, has only been demonstrated in recent years and their relevance is still questionable [9,25,26]. The GM composition is dominated by mostly obligate anaerobic bacteria, with the main phyla being Firmicutes and Bacteroidetes, while others, such as Actinobacteria and Proteobacteria, are represented to a lesser extent.

In contrast, the airways harbor a distinct microbial ecosystem, featured by less diversity and richness and composed mainly of the genera Prevotella, Streptococcus, Veillonella, Fusobacterium, and Haemophilus [24,27]. Gut and lung microbial communities are known to influence each other during infections in an interesting, still not completely understood, crosstalk called the “gut-lung axis” (Table 1).

### 2.1. Innate Immunity

In the gut–lung axis, the key to the interaction could potentially be the many metabolites produced and/or influenced by GM, which have been shown to have a regulatory function of the immune system.

Among these, short-chain fatty acids (SCFAs), which include acetate, propionate, and butyrate, are end-products of the fermentation of fibers by commensal bacteria in the cecum and the colon. SCFAs not used for energy by intestinal epithelial cells are released into the circulation and reach distant sites, such as adipose tissue, liver, pancreas, lungs, brain, and bone marrow [22,24,33]. Butyrate administration has been shown to enhance the response to influenza infection, both by stimulating the production of macrophage precursors and by downregulating the damage due to the infiltration of neutrophils [24,28]. Trompette et al. demonstrated in mouse models that a high-fiber diet (HFD), resulting in the production of SCFAs in the gastrointestinal tract, is protective against influenza through two mechanisms described below. HFD mice showed enhanced bone marrow hematopoiesis of Ly6c-patrolling monocytes, which mature in macrophages with a limited ability to generate chemokine ligand 1 (CXCL1) chemokines in the lungs. The lower presence of CXCL1 resulted in a reduced recruitment of neutrophils to the airways and a subsequent lower release of mediators responsible for the damage in the lung tissue. Simultaneously, SCFAs boosted the function of CD8+ T cells that displayed an increased ability to kill virus-infected cells [28]. It should also be remembered that SCFAs are involved in the activation of G protein-coupled receptors (GPR40-43, also known respectively as free fatty acid receptors (FFAR1-3) [29,30,34]), which are necessary for the resolution of the inflammatory response, as seen in studies on GPR43-deficient mice [30,35]. In particular, Maslowski et al. demonstrated that in GPR43-deficient mice, SCFAs did not induce either the release of reactive oxygen species (ROS) from neutrophils or the enhancement of their phagocytic activity, as was instead noted in GPR43 mice.

Type I interferons (IFNs) are pleiotropic cytokines involved in another important signaling pathway in viral immunity. Increasing evidence shows that GM can regulate host immune homeostasis, as well as the response to injury and bacterial infections, through type I IFN signaling. In this regard, Antunes et al. studied the effects of acetate pre-treatment in RSV-infected mice. Acetate has been shown to modulate IFN-β production through type 1 IFN receptor engagement during RSV infection both in human pulmonary cells in vitro and in vivo in the lower airways of infected mice. This effect was not observed in the absence of the type 1 IFN receptor. Mice pre-treated with acetate showed an undetectable lung viral load, a reduced cell number in the bronchoalveolar lavage, and a decrease in inflammatory cells in the lungs. 

Recently, Antunes et al. also demonstrated in animal models that pre-treatment with acetate led to faster recoveries from RSV infection by enhancing the expression of retinoic acid-inducible gene I (RIG-I) and interferon-stimulated genes. In RIG-I knockout mice these protective effects were not noted, proving that RIG-I plays a role in the recovery from RSV infections. Moreover, the high stool level of acetate was significantly associated with less severe bronchiolitis, higher oxygen saturation, and diminished lasting fever [27].

Similar results were found by administering propionate and butyrate, confirming the potential prophylactic role of SCFAs in RSV infection.

Desaminotyrosine (DAT) has also been shown to be a metabolite involved in modulating the immune response. DAT derives from flavonoids, a group of polyphenolic compounds enriched in certain foods, such as tea, citrus fruits and juices, and in wine [36]. The obligate anaerobe *Clostridium orbiscindens* is responsible for the metabolism of flavonoids with the consequent production of DAT. DAT protects against influenza through the amplification of type I IFN signaling [15] and is an essential mediator of phagocytes in the lung. Without DAT, the influenza virus causes inflammation and severe disease. GM-derived metabolites therefore have the potential to modulate the innate immune response by increasing the lung production of type I IFN. These findings imply that prior colonization by specific bacteria and a flavonoid-enriched diet are both key components that modulate immune response to influenza infection.

There are also other known immunomodulatory gut metabolites, such as indole derivatives, niacin, urolithin A, and pyruvic and lactic acids, which may affect respiratory health, but their role is not yet understood [24].

In addition, GM commensal bacteria appear to be able to directly prime innate immunity in response to an influenza infection in the lungs. Ichinohe et al., for example, showed that GM composition can regulate virus-specific CD4+ and CD8+ T cell generation and antibody responses following respiratory influenza virus infection [16]. Oral antibiotic treatment resulted in defective CD4+ T, CD8+ T, and B-cell immunity following intranasal infection with influenza virus, while distal (rectal) inoculation of Toll-like receptor (TLR) agonists (TLR9 and TLR3 agonists) could rehabilitate the pulmonary immune reaction to infection in antibiotic-treated mice. GM is also involved in the stimulation of the transcription and translation of pro-interleukin-1beta (ILs) and in the activation of the caspase that transforms pro-IL1beta into the mature one. In the study of Ichinohe et al., IL1beta levels in bronchoalveolar lavage were altered in influenza-infected mice that were treated with antibiotics. Therefore, GM integrity (i.e., maintenance of a eubiotic configuration) is required to maintain adequate pro-IL1beta expression, while antibiotic treatment for respiratory infections could have a deleterious effect on the immune response to the influenza virus.

Moreover, the integrity of GM results to be necessary for the correct function of dendritic cells (DCs). Negi et al. found that antibiotics’ treatment reduced the expression of macrophage-inducible C-type lectin (mincle) in lung DCs during *Mycobacterium tuberculosis* (Mtb) infections [37]. Specifically, DCs that expressed a lower presence of mincle were less capable of activating naïve CD 4 T cells with the result of the increased survival of Mtb. In vivo, the administration of mincle ligan restored the immune defect of lung DCs [37].

### 2.2. Adaptative Immunity

GM is also involved in adaptative immunity, enhancing the function of CD8+ T cells that play a key role against viral infections [16,28,31]. In particular, a greater availability of glucose, derived from the catabolic processes of GM, stimulates the effector function of these cells by increasing the differentiation and activation process [28,38].

Another piece of evidence of GM involvement comes from in vivo models with antibiotic exposure. Thackray et al. have shown that antibiotic-treated mice have an impaired response to flavivirus infection due to a decreased number of specific CD8+ T cells in the spleen and popliteal lymph nodes. Similarly, Ichinohe et al. showed reduced numbers of not only CD8+ T, but also of CD4+ T and B-cells in mice treated with oral antibiotics [16,31].

The role of GM in modulating the severity of RSV infection has been also demonstrated in animal models. Fonseca et al. observed that oral *Lactobacillus johnsonii* supplementation in mice led to an attenuated immune response to RSV. This effect was mediated by the recirculation of systemic metabolites including docosahexanoic acid, decreasing airway Th2 cytokines, dendritic cell function, and improving T-regulatory cells [39].

The gut–lung axis also appears to operate through the direct movement of cells from the gut to the respiratory system [24]. This migration of immune cells was demonstrated in the study by Huang et al., where, after connecting the circulation of two mice, labeled type 2 innate lymphoid cells (ILC2s) from a mouse were found in the lungs of both mice [40]. This did not occur in mice treated with antibiotics and contaminated with the gastrointestinal nematode Nippostrongylus brasiliensis [25], thus suggesting the relevance of an intact GM.

There is also evidence of a crucial role for GM against lung bacterial infections [32,41]. Brown et al. highlighted that GM can protect against *Streptococcus pneumoniae* and *Klebsiella pneumoniae* infections. To prove this, mice were treated with antibiotics and subsequently infected with the bacteria. In treated mice, increased lung bacterial counts and a diminished pulmonary production of granulocyte–macrophage colony-stimulating factor (GM-CSF) and chemokine ligands 1 and 2 (CXCL1 and CXCL2) were noted. To demonstrate the role of GM-CSF in the response to respiratory infections, mice were given GM-CSF-neutralizing antibodies, and this caused impaired alveolar macrophage development. Defects in bacterial clearance could be restored in antibiotic-treated mice by administering recombinant GM-CSF but not neutralizing CXCL1 and CXCL2. Moreover, antibiotic-induced GM disruption resulted in secondary IgA deficiency in both mice and humans, which led to an enhanced susceptibility to *Pseudomonas aeruginosa*-induced pneumonia [24]. Another confirmation of the important role of GM in respiratory bacterial infections comes from the study by Gauguet et al., who showed that the segmented filamentous bacteria present in GM play a role in defending against *Staphylococcus aureus* pneumonia by stimulating the production of the Th17 cytokine, IL22, and by enhancing the number of neutrophils in the lungs [42]. 

This fascinating immunological crosstalk along the gut–lung axis has not yet been fully explored, but the already known mechanisms underlying these interactions could be used in the near future for novel evidence-based strategies for the prevention and treatment of RTIs (Figure 1).

## 3. GM and RTIs in Children

We conducted a narrative review of the literature describing the studies dealing with the relationship between GM and RTIs in the pediatric population (Table 2). Methods of the research are attached in the Appendix A.

In 2016, Hasegawa et al. analysed for the first time the correlation between RTIs and GM in children [43]. Specifically, they set up a case-control study where they collected stool samples from 40 hospitalized children with bronchiolitis and 115 healthy controls. By 16S rRNA gene sequencing, they stratified all the GM profiles into four groups based on the most represented genera: *Escherichia* (30%), *Bifidobacterium* (21%), *Bacteroides* (28%), and *Enterobacter*/*Veillonella* (22%). Then, they analysed the proportion of severe bronchiolitis in the four groups, finding the highest incidence in the *Bacteroides*-dominant profile (44%) and the lowest in the *Enterobacter*/*Veillonella*-dominant profile (15%) (odds ratio (OR) 4.95; 95% confidence interval (CI) 1.58–15.5; *p* = 0.008). In that paper, for the first time, an association between the GM composition and respiratory infections in infants was demonstrated.

Recently, Harding and colleagues profiled GM (by 16S rRNA gene sequencing) in RSV-infected infants, finding for the first time a relationship with bronchiolitis severity. The study was conducted on 95 infants, including 37 healthy, 53 hospitalized for moderate RSV bronchiolitis (general ward), and 5 with severe bronchiolitis (requiring paediatric intensive care unit hospitalization). Stool samples were collected within 72 h of admission. According to their results, RSV-infected patients and controls segregated by β-diversity (i.e., inter-individual diversity), with higher proportions of *Muribaculacae* (*S24_7*), Clostridiales, *Odoribacteraceae, Lactobacillaceae*, and *Actinomyces* in the former. Moreover, there were differences in GM composition based on RSV disease severity, with higher relative abundances of *S24_7* in severe patients than moderate patients and the controls. However, as expected, it was unknown whether these changes in GM were causal or caused by RSV infection [44].

Alba et al. in 2021 conducted a case-control study aimed at evaluating GM and nasal microbiota during bronchiolitis. They collected stool and nasal samples from 78 hospitalized children diagnosed with bronchiolitis and 17 healthy children as controls, which underwent 16S rRNA gene sequencing. Regarding GM, they did not find differences between the study groups, but concluded that the small number of study participants, especially in relation to the healthy controls, may have biased the results, preventing the identification of significant differences [45].

In recent years, many papers have described the GM composition in newborns in relation to various risk factors, but the association with RTIs in the first years of life has not been well established. Reyman et al. in 2019 designed a study to evaluate GM in 74 vaginally delivered and 46 caesarean section-born children. Significant differences were found in the GM composition, especially in the first week of life, regardless of intrapartum antibiotic administration. Consistent with the literature, vaginally delivered children showed overrepresentation of *Bifidobacterium* spp. and reduced abundances of *Klebsiella* spp. and *Enterococcus* compared to children born by caesarean section. As a secondary endpoint, the authors also assessed whether GM composition was associated with health outcomes, specifically RTIs and antibiotic treatment during the first year of life. According to their findings, a prevalence of *Bifidobacterium* spp. in the first week of life was significantly associated with fewer RTI events, whereas *Klebsiella* spp. and *Enterococcus* prevalence were negatively associated. This result confirms the impact of GM on health since the first days of life, even though these data should be confirmed in a larger cohort and specifically designed studies [12].

Li et al. in 2019 profiled GM in 26 children with RRTIs and 23 healthy volunteers. In the RRTI group, α-diversity (i.e., intra-sample diversity—a hallmark of healthy GM) was significantly lower. A clear separation in β-diversity was also observed between the study groups, suggesting that respiratory disease is an important factor accounting for GM changes. While the predominant phyla in both groups were *Firmicutes* and *Bacteroidetes*, a clear underrepresentation of *Verrucomicrobia* and *Tenericutes* was found in the RRTI group. At the genus level, the main differences included an overrepresentation of *Enterococcus* and a concomitant decrease in typically health-associated taxa, such as *Eubacterium*, *Faecalibacterium*, and *Bifidobacterium*, in the RRTI group. Using a random forest model, the authors sought to identify GM biomarkers that could discriminate RTTI status. *Enterococcus* was found to be the best performing taxon in RRTI identification, followed by *Eubacterium* [19].

Li et al. in 2019 also designed a study to assess the role of GM dysbiosis and probiotic intervention in RRTIs. For what concerns GM dysbiosis, a total of 90 RRTI children, divided into an “active group” (with respiratory symptoms or with symptom resolution within 3 days) and a “remission group” (without history of respiratory infections for more than one week), were compared with a control group of 30 healthy children. Stool samples were collected and analysed through 16S rRNA gene sequencing. In this study, the number of lactobacilli and bifidobacteria was significantly reduced in children with RRTIs, both in the active and in the remission group, compared to the control group [46].

A recent study investigated the association between GM dysbiosis and SARS-CoV-2 infection. Xu et al. collected throat swabs, nasal swabs, and faeces from 9 children affected by COVID-19 (aged 7 to 139 months) and compared them with 14 age-matched healthy control children. For what concerns the GM profile, a significantly higher evenness was observed in COVID-19 children than in the controls, with increased representation of *Bacteroidetes* and *Firmicutes*, and a decrease in *Proteobacteria*. In general, typical commensal bacteria were significantly more abundant in the gut of the controls, whereas COVID-19 children were enriched in opportunistic pathogenic and environmental bacteria, in particular *Pseudomonas, Herbaspirillum*, and *Burkholderia.* Patients were followed up for 25–58 days after symptom onset to reconstruct GM dynamics over time. Improvement and/or restoration of GM composition was observed in three COVID-19 children, while further deterioration occurred in three other children. These changes showed no association with clinical recovery or the presence of SARS-CoV-2 RNA in the gut. This research showed that SARS-CoV-2 infection is also associated with GM dysbiosis in children, similar to other viral RTIs we have already described [47].

One of the largest and most ambitious studies on the role of the gut–lung axis in respiratory disease is the ongoing German multicentre cohort study LoewenKids. Since 2015, 782 newborns have been enrolled, with the aim of obtaining from every child information about the complete history of the infection, in combination with information on the development of GM and nasal microbiota, the genetic background, and the child’s environment. Symptom information using daily diaries filled in by parents is also being collected, and nasal or stool swabbing is performed whenever a respiratory or gastrointestinal symptom is referred. Nasal, stool, and blood samples are also collected every 2–4 times a year in asymptomatic children [48]. This study could probably help clarify the role of GM in immune system development and its correlation with respiratory infections over the next few years.

## 4. Conclusions

The gut–lung axis and, in particular, the relationship between GM profile and RTIs have been thoroughly investigated in animal models in recent years. Several pathways have been described by which GM could directly or indirectly modulate the immune system response to infections. However, the impact of GM in humans and especially children has not yet been fully established. Thus, a potential translation from bench to bedside of this growing knowledge is far to be reached.

Moreover, children represent a peculiar population in which the GM has not reached its final maturation yet and this may provide more possibility of targeted interventions to change its composition [49,50].

The human studies we exposed (Table 2) mainly describe an association between GM dysbiosis and RTIs. Three of them [34,44,45] show a relation between GM composition and bronchiolitis infection in infants, whereas two others [19,46] observe a difference in GM composition in RRTIs compared to healthy controls. As stated by Hasegawa et al. [43], this correlation between GM dysbiosis and RTIs suggests a causal pathway. In addition, the preclinical data discussed above seem to suggest that there may be a causative link between GM composition and the respiratory (innate and adaptive) immune response against pathogens [11,22]. On the other hand, the hypothesis that GM alterations might be a marker of greater propensity to RTIs or possibly, that viral infections alter the gut microenvironment, leading to dysbiosis, cannot be entirely excluded.

Modifications in GM homeostasis could therefore impact the risk of developing a respiratory infection and also its severity. This is of particular interest because, if confirmed, the GM composition could be used as a possible parameter to assess the risk of progression of RTIs. Reyman et al. [12] further underlined the role of GM dysbiosis in the first days of life, with a potential impact on the future risk of developing RTIs. This is another observation highlighting the importance of GM from birth, and particularly in the first months of life, for the future health of the child and adult. Nevertheless, it should be noted that there is still no clear identification of the genera most associated with RTIs (Table 2), which makes it difficult to draw definitive conclusions.

Although mounting evidence shows a correlation between GM and RTIs in childhood, the different target populations analysed, the study design, the small number of participants, and the presence of confounding factors could explain why a unique GM signature could not be found. These limits need to be overcome in the future by analysing larger cohorts. In our opinion, a better understanding of the relationship between GM and RTIs must be sought and, once achieved, could potentially lead to a new approach to the prevention and treatment of one of the most common diseases of childhood, namely RTIs in infants and children.

## Figures and Tables

**Figure 1 nutrients-14-02992-f001:**
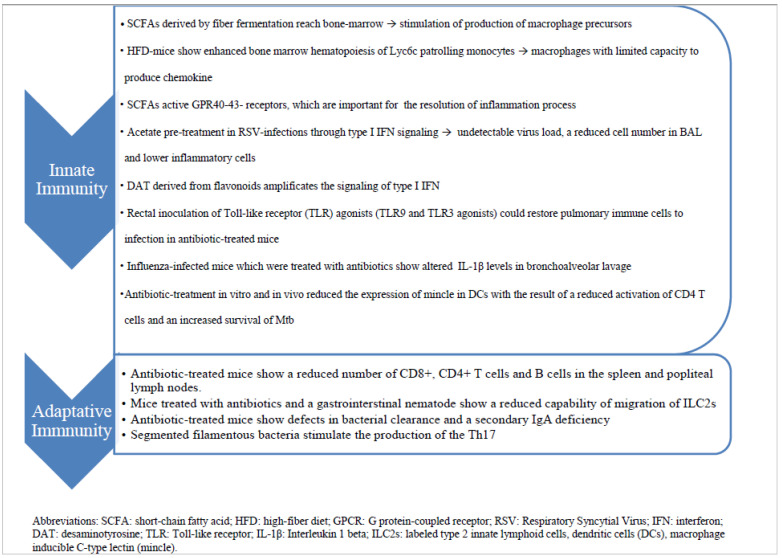
The diagram summarizes the immune mechanisms by which gut–lung axis works and potentially influences the answer to infections.

**Table 1 nutrients-14-02992-t001:** Studies exploring the mechanisms through which the gut microbiota interacts with the lungs and affects the response to infections.

Study, Year	Aim	Results
Trompette et al., 2018 [28]	Effect of an HFD, through SCFA modulation in influenza-infected mice.	HFD in murine models is protective against influenza affecting bone marrow hematopoiesis by shaping alternative macrophages that produce less CXCL1chemokine, resulting in reduced neutrophil recruitment and tissue damage. SCFAs-boosted CD8+ T cell effector function by enhancing cellular metabolism.
McAleer et al., 2017 [29]	Effects of GM composition on lung immunity.	HFD can increase the prevalence of Bacteroidetes and Actinobacteria members, as well as the production of SCFAs. Dysbiosis resulting from dietary fat or antibiotic use enhances lung inflammation in response to allergens or infections. Antibiotic use may inhibit the phagocytic capacity of alveolar macrophages, increasing the susceptibility to opportunistic infections in the lungs.
Maslowski et al., 2009 [30]	The role of SCFAs in the regulation of the immune response by GPR43 activation.	Stimulation of GPR43 by SCFAs is necessary for the resolution of inflammatory responses. GPR43-deficient (Gpr43−/−) mice showed exacerbated or unresolved inflammation in models of colitis, arthritis, and asthma. This appeared to be related to the increased production of inflammatory mediators by Gpr43−/− immune cells and increased immune cell recruitment, and the same occurred in germ-free mice. GPR43 binding of SCFAs potentially provides a molecular link between diet, GM metabolism, and immune and inflammatory responses.
Antunes et al., 2019 [27]	The role of SCFAs in RSV infection.	Acetate administration-mediated IFN-β response, resulting in a reduction of viral load and cell count in bronchoalveolar lavage, and a reduction of inflammatory cells in the lungs of RSV-infected mice. Type 1 IFN signaling via the IFN-1 receptor is essential for acetate antiviral activity in pulmonary epithelial cell lines and for an acetate protective effect. GPR43 activation in pulmonary epithelial cells reduced virus-induced cytotoxicity and promoted antiviral effects through IFN-β response; this was not reported in Gpr43−/− mice.
Steed et al., 2017, [15]	Evaluation of DAT in protecting from influenza through type I IFN.	DAT, a microbiol metabolite, protects against influenza through the augmentation of type I IFN signaling and the decrease in lung immunopathology. A specific human-associated gut microbe, *Clostridium orbiscindens*, produced DAT and rescued antibiotic-treated influenza-infected mice.
Ichinohe et al., 2011 [16]	The function of GM in influenza A-infected mice.	Oral antibiotic treatments resulted in defective CD4+ T-, CD8+ T-, and B-cell immunity following intranasal influenza A infection in mice. Injection of TLR ligands appeared to rescue immune impairment in these mice. GM is involved in stimulating the transcription and translation of pro-IL1beta and in the activation of the caspase that transforms pro-IL1beta into the mature form.
Thackray et al., 2018 [31]	Effects of oral antibiotics in flaviviridae-infected mice.	Oral antibiotics treatment in flaviviridae infections has multiple effects: ❖ Increases viral load; ❖ Impairs virus-specific CD8+ T cell responses; ❖ Raises the risk of severe disease.
Wypich et al., 2019 [24]	Analysis of the gut–lung axis and its communication pathways.	SCFAs influence immune cell development in the bone marrow. Then, bone marrow-derived cells shape immune responses in distal body sites, such as the lungs. Cells migrating from the gut to the lungs may influence respiratory immunity, i.e., ILC2s, ILC3s. and TH17 cells migrating from the gut into the lungs. The microbial metabolite DAT protects the host against influenza virus infection via the augmentation of type I IFN responses.
Brown et al., 2017 [32]	GM signaling that protects against respiratory infections.	Models of commensal colonization in antibiotic-treated and germ-free mice, using cultured commensals from the Bacteroidetes, Firmicutes, Actinobacteria, and Proteobacteria phyla show that these bacteria have the ability to stimulate Nod2. GM enhances respiratory defenses via GM-CSF signaling, which stimulates pathogen killing and clearance by alveolar macrophages. Potent Nod-like receptor(NLR)-stimulating bacteria in GM (*Lactobacillus reuteri*, *Enterococcus faecalis*, *Lactobacillus crispatus*, and *Clostridium orbiscindens*) promote resistance to lung infections through Nod2 and GM-CSF.

DAT: desaminotyrosine; GPCR: G protein-coupled receptor; GM: gut microbiota; GM-CSF: granulocyte–macrophage colony-stimulating factor; HFD: high-fiber diet; IL: interleukin; ILC: innate lymphoid cell; IFN: interferon; SCFA: short-chain fatty acid; TLR: Toll-like receptor, NLR: Nod-like receptor.

**Table 2 nutrients-14-02992-t002:** Studies analysing the relationship between RTIs and GM in children.

Author, Year of Publication, Country	Study Design	No. of Subjects and Population	Age Range	Aim	Main Results
Hasegawa et al. [43], 2016, Massachusetts	Case-control	40 hospitalized infants with bronchiolitis vs. 115 healthy controls.	<12 months	To identify faecal microbiota profiles and compare the likelihood of bronchiolitis.	The highest likelihood of bronchiolitis in the *Bacteroides*-dominant profile compared to the *Enterobacter/Veillonella*-, *Escherichia*- and *Bifidobacterium*-dominant profiles.
Harding et al. [44], 2020, Louisiana	Case-control	53 hospitalized infants with RSV- bronchiolitis vs. 37 healthy controls.	<7 months	To compare GM in children with different bronchiolitis severity vs. controls.	Increase in *S24_7*, Clostridiales, *Odoribacteraceae*, *Lactobacillaceae*, and *Actinomyces* in patients with bronchiolitis compared to controls. Increase in *S24_7* in severe patients compared to moderate patients and controls.
Alba et al. [45], 2021, Spain	Case-control	58 infants with RSV-bronchiolitis vs. 17 healthy controls.	<24 months	To compare GM in children with bronchiolitis vs. controls.	No significant differences regarding the most abundant genera (*Bifidobacterium*, *Streptococcus*, and *Escherichia*) between infants with bronchiolitis and controls.
Reyman et al. [12], 2019, Netherlands	Prospective single centre	74 VD children, 46 born by CS.	First year of life	Differences in GM between VD and CS-born children. Correlation between GM and RTIs in the first year of life.	Prevalence of *Bifidobacterium* in the first week of life was significantly associated with fewer RTI events. *Klebsiella* and *Enterococcus* prevalence were negatively associated with fewer RTI events.
Li et al. [19], 2019, China	Case-control	26 children with RRTIs vs. 23 healthy controls.	>5 years	To compare GM in children with RRTIs vs. controls. To identify GM biomarkers that could discriminate RRTI status.	Alpha diversity in the RRTI patients’ GM was significantly lower. Reduction of Verrucomicrobia and Tenericutes phyla with increase in *Enterococcus* and decrease in *Eubacterium* in the RRTI group. *Enterococcus*, taken as a biomarker, showed the highest accuracy in identifying RRTIs.
Li et al. [46], 2019, China	Case-control	90 children with RRTIs vs. 30 heathy controls.	<11 years	To compare GM in children with RRTIs vs. controls. Effects of probiotics on GM and RRTIs.	Significant reduction of lactobacilli and bifidobacteria in children with RRTIs.
Xu et al. [47], 2021, China	Case-control	9 children with COVID-19 vs. 14 healthy controls.	<12 years	To compare GM in children with COVID-19 vs. controls.	Increased representation of Bacteroidetes and Firmicutes, and decrease in Proteobacteria, with significant increase in opportunistic pathogenic and environmental bacteria in COVID-19 children.

COVID-19: coronavirus disease 2019; CS: cesarean section; GM: gut microbiota; RRTI: recurrent respiratory tract infection; RTI: respiratory tract infection; VD: vaginally delivered.

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
