# Peer review of "The Relationship between Gut Microbiota and Respiratory Tract Infections in Childhood: A Narrative Review"

_nutrients, 2022, doi:10.3390/nu14142992_

Round 1
Reviewer 1 Report
Zama D et al. reviewed the relationship between gut microbiota and respiratory tract infections in childhood. This narrative review is well-written and informative. But it would be better to include the instructions for how to study the complex interactions between GM and infection disease.
Specific comment:
1) The first sentence of the abstract is incomplete
2) Figure 1 is missing
Author Response
July 2022
Dear Editor-in-chief,
thank you very much for having reviewed our manuscript entitled "The relationship between gut microbiota and respiratory tract infections in childhood: a narrative review" that we had submitted for consideration for Nutrients (Manuscript ID: nutrients-1800245).
We are very grateful for the Reviewer’s very constructive comments that much improved the quality of the paper.
Please find attached an itemized point-by-point response to all the comments raised by the Reviewers.
Thanking in advance for your attention and looking forward to hearing from you at your convenience,
Kind regards,
Lorenzo Biscardi, MD
Reviewer reports:
Reviewer #1: Zama D et al. reviewed the relationship between gut microbiota and respiratory tract infections in childhood. This narrative review is well-written and informative. But it would be better to include the instructions for how to study the complex interactions between GM and infection disease.
Specific comment:
1) The first sentence of the abstract is incomplete
We would like to thank Reviewer #1 for the constructive comments. Firstly, we would like to apologize for having forgot during the submitting phase to insert the complete abstract. We now provided to also insert the missing sentence.
2) Figure 1 is missing
We would like to thank Reviewer #1 for the report. We uploaded Figure 1 as a complementary file because of its dimension. We will provide to make sure that it is possible to download it.
To also answer the previous observation, we believe that Figure 1 tries to summarize and explain most of the mechanisms of interaction between gut microbiota and immune system and therefore we hope it could be helpful in this contest.
Reviewer #2: The authors of this article provide a narrative review of research findings on gut microbiota and respiratory tract infections in children. Current studies on the homeostasis of gut microbiota and its metabolites have shown that its functions are diverse. This paper mainly focuses on the achievements of gut microbiota and immunity. The summary and analysis of it is of great significance for fully understanding the gut-lung axis.
The main problem of this article was that the abstract part was incomplete, please revise it.
We would like to thank Reviewer #2 for the constructive comments and kind words. We would like to apologize for having forgot during the submitting phase to insert the complete abstract. We now provided to also insert the missing sentence.
Reviewer 2 Report
The authors of this article provide a narrative review of research findings on gut microbiota and respiratory tract infections in children. Current studies on the homeostasis of gut microbiota and its metabolites have shown that its functions are diverse. This paper mainly focuses on the achievements of gut microbiota and immunity. The summary and analysis of it is of great significance for fully understanding the gut-lung axis.
The main problem of this article was that the abstract part was incomplete, please revise it.
Author Response

(The authors gave the same response as above.)
